# High-Fat Diet and Metabolic Diseases: A Comparative Analysis of Sex-Dependent Responses and Mechanisms

**DOI:** 10.3390/ijms26104777

**Published:** 2025-05-16

**Authors:** Qiaoling Mo, Xinquan Deng, Ziyu Zhou, Lijun Yin

**Affiliations:** School of Sports, Shenzhen University, Shenzhen 518060, China; 2400201012@mails.szu.edu.cn (Q.M.); 2023331017@email.szu.edu.cn (X.D.); 2023332078@email.szu.edu.cn (Z.Z.)

**Keywords:** high-fat diet, sex difference, metabolic diseases, mechanism

## Abstract

Sex differences in metabolic disorders and susceptibility to chronic diseases induced by a high-fat diet (HFD) exhibit significant dimorphic characteristics. A long-standing male-centric bias in medical research and healthcare, predominantly focused on male physiological traits, has hindered the precise treatment of metabolic diseases in female patients. A comprehensive understanding of sex differences in metabolic health and their underlying mechanisms is crucial for advancing personalized health promotion and precision medicine. This review systematically elucidates sex-specific manifestations in high-fat diet-associated metabolic disorders: males predominantly develop visceral adiposity, insulin resistance, and dyslipidemia, accompanied by a significantly elevated risk of cardiovascular and metabolic syndromes. Premenopausal females maintain metabolic homeostasis through the estrogen-mediated optimization of glucose and lipid metabolism and oxidative stress buffering mechanisms, whereas postmenopausal-phase females experience dramatic metabolic vulnerability due to z loss of protective barriers. Furthermore, we emphasize multidimensional mechanistic interpretations of metabolic sexual dimorphism from perspectives including sex chromosome complement, sex hormone signaling pathways, epigenetic regulation, gut microbiota composition, and neuroendocrine dimorphism. This work provides critical theoretical foundations for rectifying unisex research paradigms and optimizing sex-specific early warning systems and precision therapeutic strategies for metabolic disorders.

## 1. Introduction

Global economic expansion and aggressive food marketing strategies have precipitated profound shifts in dietary patterns worldwide. Notably, the consumption of energy-dense foods rich in saturated fatty acids, trans-fatty acids, and ultra-processed ingredients has surged across developed nations and rapidly urbanizing low-to-middle-income countries [1,2,3,4]. Chronic HFD exposure, particularly involving saturated fatty acids and trans-fatty acids, has been mechanistically linked to escalating epidemics of obesity [5], cardiometabolic diseases, diabetes, and other many metabolic diseases, which constitutes a paramount global public health challenge [6]. Mechanistically, excessive fat intake dysregulates lipid homeostasis by elevating serum low-density lipoprotein cholesterol (LDL-C) while suppressing high-density lipoprotein cholesterol (HDL-C), thereby accelerating atherogenesis and cardiovascular events [7]. In parallel, HFD induces pancreatic β-cell dysfunction and systemic insulin resistance (IR), establishing a causal pathway to type 2 diabetes mellitus (T2DM) [2]. Furthermore, HFD potentiates inflammation through NF-κB (nuclear factor-kappa B) pathway activation and reactive oxygen species (ROS) production [8,9].

Emerging evidence reveals striking sexual dimorphism in HFD-associated metabolic responses. Preclinical models demonstrate male-predominant susceptibility, with accelerated adiposity gain, adipose tissue inflammation, macrophage infiltration [10], and IR [11], contrasting with relative protection in females during reproductive age [12,13]. Clinical observations, though limited, corroborate this sex-divergent vulnerability: males exhibit higher metabolic syndrome prevalence, while premenopausal females maintain protection through estrogen-mediated mechanisms. Postmenopausal estrogen decline abolishes this advantage, rendering females vulnerable to cardiovascular dysfunction [14] and IR [15]. Sexual dichotomy extends to IR subtypes—impaired glucose tolerance (postprandial phase) predominates in females versus impaired fasting glucose in males [16,17]. Alarmingly, the persistent male-centric bias in biomedical research compromises clinical practice. Current diagnostic frameworks and therapeutic protocols predominantly derive from male-predominant studies [18], exemplified by the U.S. Government Accountability Office report identifying 80% of withdrawn drugs (1997–2000) as posing elevated risks for females [19]. This underscores the imperative to integrate sex as a critical biological variable in research design.

This review synthesizes current evidence on HFD-induced metabolic dysregulation through a sex-specific perspective, elucidating underlying mechanisms spanning molecular pathways to clinical manifestations. By bridging this knowledge gap, we aim to catalyze sex-informed research paradigms, advance personalized therapeutic strategies, and propel precision medicine for metabolic disease management.

## 2. Sex Differences in Metabolic Responses Associated with HFD and the Development of Metabolic Diseases

Chronic HFD consumption serves as a critical etiological factor driving pathological alterations in body composition, metabolic homeostasis, inflammatory cascades, and oxidative stress profiles, which collectively establish the pathophysiological foundation for diverse metabolic disorders. This review systematically examines the sexually dimorphic responses to HFD exposure and differential progression trajectories of obesity, diabetes, and cardiovascular diseases between sexes through multidimensional analysis encompassing these pathological alterations (Figure 1). To achieve this goal, a systematic literature search was conducted across PubMed, Web of Science, and Science Direct using MeSH-based keywords (e.g., “high-fat diet”, “sex differences”, “metabolic diseases”, “gut microbiota”) combined with Boolean operators (e.g., “HFD AND sex differences AND mechanisms”). Inclusion criteria encompassed English-language preclinical and clinical studies exploring sex-dependent mechanisms. The exclusion criteria applied were (i) studies with an off-topic or non-mechanistic focus and (ii) gray literature (Ph.D. dissertations, conference proceedings) or unpublished data. Reference lists of key articles were manually screened to ensure comprehensive coverage of sex-specific molecular pathways and therapeutic interventions.

Prior to mechanistic dissection, conceptual clarification remains imperative. The term “sex differences” denotes biologically determined disparities rooted in chromosomal complement, gonadal hormone profiles, and reproductive system architecture. In contrast, “gender differences” encompass sociocultural constructs shaped by psychosocial roles, behavioral norms, and identity expressions within specific cultural contexts [20]. While acknowledging potential gender-related influences, this review focuses explicitly on biological sex-specific metabolic adaptations, particularly those governed by physiological determinants.

### 2.1. Sex Differences in Body Composition Abnormalities and Obesity Development

Sexual dimorphism in body composition and adipose tissue distribution is well documented, with males typically exhibiting greater lean mass, higher basal metabolic rates, and lower overall adiposity compared to females [21,22]. This divergence extends to fat deposition patterns: males predominantly accumulate visceral fat within abdominal compartments, whereas females demonstrate preferential subcutaneous fat storage in gluteofemoral regions. Such anatomical partitioning aligns with sex-specific energy allocation strategies. Premenopausal females preferentially utilize carbohydrates and lipids for energy production, a metabolic orientation that promotes subcutaneous fat sequestration while mitigating visceral and ectopic lipid deposition [20]. At the cellular level, male visceral adipocytes exhibit heightened lipid storage capacity under HFD conditions [23] and demonstrate elevated secretion of proinflammatory cytokines [24]. These inflammatory mediators disrupt insulin signaling cascades [25], establishing a mechanistic link to the increased prevalence of metabolic syndrome (MetS) observed in males. MetS is a cluster of metabolic risk factors that include central obesity, hypertension, insulin resistance, and atherogenic dyslipidemia [26]. It serves as a critical precursor to life-threatening comorbidities, elevating risks for cardiovascular diseases and T2DM [22]. Conversely, female subcutaneous adipose tissue displays distinct metabolic properties, including enhanced insulin sensitivity [20,27]. Clinical evidence reveals accelerated weight gain trajectories in male versus female subjects under HFD conditions, a pattern consistently replicated in rodent models [28]. These findings collectively support the clinical consensus that males exhibit greater susceptibility to central obesity-associated metabolic disorders compared to premenopausal females.

However, the decline in estrogen levels during menopause drives a masculinized shift in fat distribution, characterized by preferential visceral adipose tissue accumulation, which increases the risk of obesity and other metabolic diseases. A recent clinical study on 3.5 million Chinese older adults reported that the prevalence of central obesity was higher in older females than in males (37.4% vs. 25.2%) [22]. This suggests that estrogen plays a critical role in regulating lipolysis metabolism in females [29]. For instance, estrogen is regulated by the expression of aromatase genes, which inhibits the hypertrophy of visceral fat depots mainly through the activation of estrogen receptor α (ERα) [29]. Additionally, the calcium-sensing receptor in adipocytes may mediate sex differences in fat distribution. HFD-fed females have lower visceral white adipose tissue mass than males, but this sex difference disappears in calcium-sensing receptor-deficient mice [30].

### 2.2. Sex Differences in Disorders of Glucose Metabolism and Development of T2DM

T2DM and its complications represent a critical global health burden, with prevalence projected to reach 12.2% by 2045 due to aging populations and urbanization [31]. The disease pathogenesis centers on two interconnected defects: IR in peripheral tissues and progressive pancreatic β-cell failure. IR refers to a defect in the control of glucose metabolism mediated by insulin, which manifests as impaired glucose uptake in the skeletal muscle, adipose tissue, and liver, coupled with enhanced hepatic gluconeogenesis, collectively elevating systemic glucose levels. Compensatory hyperinsulinemia initially mitigates hyperglycemia but ultimately drives β-cell exhaustion through endoplasmic reticulum stress and apoptotic pathways, culminating in irreversible insulin secretion defects [32]. Both preclinical and clinical evidence confirms that HFD exposure accelerates diabetes progression through coordinated mechanisms, including insulin signaling disruption [33,34], gut microbiota dysbiosis [35], ectopic lipid deposition [36], and neuroendocrine dysregulation. HFD-induced norepinephrine surges trigger uncontrolled lipolysis, resulting in systemic fatty acid overflow in rodents. This lipotoxic environment directly exacerbates IR while promoting β-cell dysfunction, establishing a self-reinforcing cycle of metabolic deterioration [37].

Sexual dimorphism in HFD-induced metabolic dysregulation is particularly evident across glycemic parameters. Clinical evidence indicates that males exhibit heightened susceptibility to fasting hyperglycemia [20], exaggerated postprandial glucose excursions [38], earlier onset of IR, and 30% lower insulin sensitivity than age-matched premenopausal females [39]. Enhanced insulin secretion capacity and delayed gastric emptying in females collectively buffer postprandial glycemic spikes under HFD conditions [40]. Longitudinal studies reveal sex-divergent HbA1c trajectories: males show greater HFD-associated HbA1c elevation, strongly correlated with oxidative stress severity [41], while premenopausal females maintain lower HbA1c through estrogen-mediated protection. This advantage dissipates after menopause, with declining estrogen levels precipitating parallel increases in fasting glucose, postprandial hyperglycemia, and HbA1c [42,43]. Compared with women of normal menopausal age, women with early menopause or late menopause may experience a large or moderate increase in the risk of T2DM, respectively [44]. Clinical observations in hyperandrogenic states like polycystic ovary syndrome (PCOS) further validate the vital role of sex hormones, as androgen excess induces male-pattern IR in females [45].

This sexual divergence stems from fundamental biological distinctions in body composition, adipose biology, and hormonal regulation. Visceral adiposity in males drives metabolic inflammation through the elevated secretion of tumor necrosis factor-α (TNF-α) and interleukin-6 (IL-6), which directly impair insulin receptor signaling [46]. Greater muscle mass paradoxically exacerbates the systemic impact of skeletal muscle IR in young men [47]. Preclinical evidence reveals that at the pancreatic level, chronic HFD accelerates β-cell failure in males via unresolved endoplasmic reticulum stress and apoptosis [48,49,50], whereas estrogen preserves β-cell function in females through ERα-mediated suppression of cellular stress pathways [51]. Beyond β-cell protection, estrogen enhances insulin sensitivity via the activation of insulin signaling effectors and mitochondrial bioenergetics, facilitating peripheral glucose utilization in female mice [52,53]. These mechanistic insights underscore the necessity for sex-specific diabetes prevention strategies, prioritizing early intervention in males and tailored approaches for postmenopausal females to optimize precision medicine outcomes.

### 2.3. Sex Differences in HFD-Induced Lipid Metabolism Disorders and Cardiovascular Disease Development

Extensive preclinical and clinical evidence demonstrates that chronic HFD exposure disrupts systemic lipid homeostasis, manifesting as ectopic lipid deposition; elevated total cholesterol (TC), triglycerides (TG), and LDL; and reduced HDL levels [54,55]. These metabolic perturbations exhibit pronounced sexual dimorphism. Males predominantly develop visceral adiposity with heightened lipolytic activity and proinflammatory cytokine secretion, predisposing them to hepatic lipid accumulation and free fatty acid spillover [42]. Clinical data reveal a male predisposition to hypertriglyceridemia and hypoalphalipoproteinemia under HFD conditions [56], whereas premenopausal females maintain favorable lipid profiles characterized by elevated HDL and reduced TC/LDL levels [57]. Preclinical studies also reveal 2–3-fold higher hepatic TG levels and severe mitochondrial dysfunction in HFD-fed male mice versus female counterparts [58]. Mechanistically, female mice exhibit attenuated upregulation of hepatic and adipose G-protein-coupled receptors under HFD treatment, mitigating lipid metabolism impairment and dyslipidemia risk [59]. Estrogen mediates this protection by enhancing adipose lipoprotein lipase (LPL) activity to promote TG hydrolysis and lipoprotein clearance [60,61], while HFD suppresses adipose LPL activity in males, exacerbating hypertriglyceridemia [62]. Postmenopausal estrogen decline abolishes these benefits, inducing male-pattern dyslipidemia [63,64,65], while estrogen replacement therapy restores lipid homeostasis [65]. However, there is controversy regarding the use of hormone replacement therapy by postmenopausal women to reduce the risk of cardiovascular diseases. Some studies suggest that hormone replacement therapy may bring benefits, while others indicate that it might increase risks of stroke and venous thromboembolism [66]. The therapy duration and timing may explain this inconsistency [67].

These HFD-induced dyslipidemias directly fuel cardiovascular pathogenesis through atherogenic mechanisms. Abundant LDL penetrates the damaged vascular endothelium, undergoing oxidation to form oxidized LDL, which is internalized by macrophages to generate foam cells—the hallmark of early atherosclerotic lesions [68,69]. Progressive lipid accumulation forms fatty streaks, while inflammatory mediators drive smooth muscle proliferation and fibrous cap formation. Plaque rupture exposes thrombogenic components, precipitating acute coronary events [70]. Sexual dimorphism permeates this pathogenic cascade: males exhibit a higher atherosclerosis risk due to elevated LDL, reduced HDL, and impaired endothelial progenitor cell function compared to premenopausal females [71,72]. Estrogen confers cardioprotection via dual mechanisms: enhancing nitric oxide-mediated endothelial function and HDL-dependent reverse cholesterol transport while suppressing LDL oxidation and vascular inflammation [73]. Clinical evidence shows that males develop unstable, rupture-prone plaques in contrast to stabilized lesions in premenopausal females [74]. The postmenopausal female loss of estrogen elevates atherosclerosis risk in females to male-equivalent levels [74]. Epidemiological analyses confirm sex-divergent outcomes in diet-related cardiovascular diseases, with males showing higher ischemic heart disease mortality from 1990 to 2021 [75].

Emerging evidence underscores the critical role of sex-specific factors in modulating heart–brain axis interactions that drive pathophysiology in cardiovascular and cerebrovascular diseases, with shared risk factors (hypertension, diabetes, dyslipidemia) and common pathways, including systemic inflammation, neuroendocrine dysfunction, and accelerated atherosclerosis [76]. Notably, diabetes and hypertension confer higher cardiovascular risks in women. Unique female reproductive events (e.g., pregnancy complications, menopause) substantially elevate stroke vulnerability, with post-stroke outcomes showing higher disability rates and depression risks compared to men. While middle-aged men demonstrate greater stroke incidence associated with premature atherosclerosis and ST elevation myocardial infarction patterns, women face diagnostic delays, therapeutic inequities, and stress-mediated heart–brain axis activation exacerbating ischemic risk. Addressing these disparities requires gender-optimized clinical trial designs and precision interventions to advance equitable cardiovascular and cerebrovascular care [76,77].

Collectively, compelling evidence reveals fundamental sex differences in HFD-induced lipid dysregulation and consequent cardiovascular and/or cerebrovascular pathologies. Men exhibit heightened susceptibility to atherogenic dyslipidemia-driven plaque progression, whereas premenopausal women retain estrogen-mediated atheroprotection until menopause yet demonstrate elevated postmenopausal stroke risk compared to age-matched males. These findings underscore the necessity for sex-stratified approaches in cardiovascular research and clinical management. Future investigations should delineate molecular drivers of sexual dimorphism to develop precision therapies countering HFD-related metabolic and cardiovascular risks.

### 2.4. Sex Differences in HFD-Induced Inflammation, Oxidative Stress, and Associated Disease Development

HFD exposure induces systemic inflammation and oxidative stress through preferential visceral adipose deposition. This metabolic–inflammatory axis drives pathological processes including myocardial injury, cardiac functional impairment, and IR via coordinated mechanisms [78].

#### 2.4.1. Inflammation

HFD exposure induces systemic low-grade inflammation through adipose tissue-derived cytokine overproduction (e.g., TNF-α, IL-6), which exacerbates metabolic dysregulation, including IR [79,80]. This inflammatory cascade is mechanistically linked to adipose macrophage dynamics: proinflammatory M1 macrophages amplify inflammation via TNF-α and IL-1β secretion, while anti-inflammatory M2 macrophages counteract through IL-10 release [81,82,83]. Visceral adipose-derived mediators propagate systemic inflammation and oxidative damage by elevating mitochondrial ROS, accelerating DNA/protein/lipid oxidation, and promoting atherosclerotic progression [9,84]. For instance, myocardial pathological hypertrophy is frequently accompanied by intensified inflammatory responses that exacerbate cardiac injury [85].

Sexual dimorphism in HFD-induced inflammatory responses is well established. It has been demonstrated in preclinical studies that reveal elevated circulating IL-6 and TNF-α levels with predominant M1 macrophage polarization under HFD conditions in male mice [11,86], whereas females demonstrate M2-dominant macrophage profiles [81,87]. Estrogen mediates this protection by suppressing M1 polarization, enhancing IL-10 production [88,89,90], and inhibiting NF-κB nuclear translocation. Although testosterone exhibits anti-inflammatory effects through NF-κB suppression, its protective capacity is attenuated by male-predominant visceral adiposity [91,92,93,94], compounded by androgen-mediated toll-like receptor 4 (TLR4)/myeloid differentiation marker 88 (MyD88) pathway activation that potentiates inflammatory signaling [95].

Cardiovascular susceptibility further demonstrates sex-specific patterns. Female mice maintain superior aortic compliance compared to males, which develop vascular stiffness and endothelial dysfunction under HFD [96]. Estrogen confers cardiovascular protection through multi-mechanistic actions: dampening inflammation and oxidative stress, promoting angiogenesis, and preserving vasodilatory capacity [97].

#### 2.4.2. Oxidative Stress

Evidence from both preclinical and clinical studies indicates that oxidative stress arises from an imbalance between ROS production (e.g., superoxide anion, hydrogen peroxide, hydroxyl radicals) and cellular antioxidant defenses, leading to macromolecular damage that drives inflammatory cascades, tissue dysfunction [98], IR [99], neurodegeneration [100], and cardiovascular pathologies [101]. Physiological redox homeostasis is maintained through coordinated enzymatic, including superoxide dismutase (SOD), glutathione peroxidase (GPx), catalase (CAT), and non-enzymatic antioxidant systems (e.g., vitamins E/C, glutathione) that neutralize ROS [102].

Chronic HFD exposure disrupts this equilibrium, with sexually dimorphic manifestations of oxidative damage. Female organisms exhibit enhanced antioxidant capacity, effectively mitigating ROS accumulation and associated tissue injury compared to males [103]. This sex-specific protection stems from elevated baseline antioxidant enzyme activities in females, mediated principally by estrogen’s redox-modulatory effects [104]. Estrogen enhances antioxidant defenses through two synergistic mechanisms: (1) upregulating cystathionine gamma-lyase expression and activating extracellular regulated kinase 1/2-NF-κB signaling to boost SOD/GPx/CAT activity [105,106,107] and (2) preserving mitochondrial integrity to suppress ROS generation [97]. Conversely, male rodents fed an HFD demonstrate elevated plasma malondialdehyde and total oxidative status, alongside reduced total antioxidant status, glutathione levels, and GPx/CAT activity [108,109]. Androgen signaling exacerbates this vulnerability by transcriptionally repressing antioxidant enzymes. For instance, testosterone downregulates SOD, CAT, and GPx expression in human endothelial cells, impairing ROS clearance [110,111,112].

In summary, chronic HFD exposure promotes cardiovascular pathogenesis through sexually dimorphic inflammatory and oxidative stress pathways. Males exhibit heightened susceptibility to ROS-driven atherosclerosis and myocardial injury compared to premenopausal females, a disparity rooted in sex hormone regulation of redox homeostasis. Estrogen serves as a critical modulator of antioxidant defenses, while androgens paradoxically amplify oxidative vulnerability despite partial anti-inflammatory activity. These findings underscore the necessity for sex-specific therapeutic strategies targeting oxidative stress in metabolic disease management.

## 3. Mechanisms Behind the Sex Differences in the Development of Metabolic Diseases

HFD exposure drives the pathogenesis of metabolic disorders, including obesity, T2DM, and atherosclerosis, by disrupting glucose and lipid homeostasis, amplifying inflammatory cascades, and compromising oxidative stress regulation. These metabolic perturbations exhibit pronounced sex-specific manifestations rooted in fundamental biological divergences. Emerging research has revealed five interconnected mechanistic determinants of sexual dimorphism: (1) sex chromosome complement-driven transcriptional regulation, (2) sex hormone receptor signaling dynamics, (3) genetic and epigenetic regulatory networks, (4) gut microbiota–host metabolic crosstalk, and (5) neuroendocrine pathway dimorphisms (Figure 2).

### 3.1. Dosage Effects and Compensatory Mechanisms of Sex Chromosome Genes

The impact of biological sex on metabolic disease susceptibility extends beyond sex hormone variations to encompass sex chromosome dosage effects and their compensatory mechanisms [113]. The inherent imbalance in X-linked gene expression between XX (female) and XY (male) configurations is partially resolved in females through random X-chromosome inactivation (XCI). However, about 15% of X-linked genes (e.g., histone demethylase KDM5C, apoptosis inhibitor XIAP) escape inactivation, exhibiting biallelic expression that drives sex-specific metabolic regulation [114,115]. Murine four-core genotype models reveal that XX chromosomal complement, independent of gonadal hormones, promotes subcutaneous adipogenesis, whereas XY predisposes individuals to visceral obesity [116]. Clinically, Klinefelter syndrome (47,XXY) is a sex chromosome aneuploidy characterized by an additional X chromosome in males. While clinical features typically include hypogonadism, infertility, gynecomastia, and learning difficulties, phenotypic expression shows marked interindividual variability. Diagnosis is confirmed through karyotype analysis [117]. The prevalence of MetS is significantly higher in patients with Klinefelter syndrome than that in the general population [118]. Clinical evidence shows that patients with Klinefelter syndrome display central adiposity, IR, and atherogenic dyslipidemia (elevated triglyceride–glucose index, reduced HDL-C) linked to X-chromosome overdosage [119], while individuals with Turner syndrome (45,XO) exhibit heightened visceral adiposity, fasting hyperglycemia, and atherosclerosis risk due to X monosomy [120].

XCI-escape genes critically mediate metabolic sexual dimorphism through adipocyte differentiation and mitochondrial bioenergetics. Lysine-specific histone demethylase 5C (KDM5C), an X-linked escape gene overexpressed in females, serves as a key obesity determinant [121]. The KDM5 histone demethylase family (KDM5A/B/C/D) exhibits sex-divergent expression, with KDM5C (X-linked) and KDM5D (Y-linked) showing reciprocal sexual dimorphism in humans and mice [122,123,124]. Female-enriched KDM5C regulates adipogenesis via histone H3K4 methylation dynamics: in white adipose tissue, it suppresses delta-like homolog 1 to promote preadipocyte maturation [125,126]; in brown fat, it activates uncoupling protein 1 (UCP1) and mitochondrial respiratory genes to enhance thermogenesis. These effects are abolished by KDM5 inhibitors like C70 [126]. Similarly, biallelic XIAP expression may protect pancreatic β-cell function, potentially attenuating diabetes progression [127,128,129], though its sex-specific roles require validation.

Y-chromosome genes equally shape metabolic disparities. Males with Y-chromosome deletions exhibit impaired insulin sensitivity [130], while preclinical and clinical evidence shows that hepatic sex-determining region Y gene upregulation in fibrotic livers activates stellate cells via platelet-derived growth factor receptor α and high-mobility group box-1 protein signaling, exacerbating lipid metabolism-associated liver pathology in mice [131]. Despite progress, critical challenges persist: the functional divergence of XCI-escape genes between humans and mice, as well as undefined sex chromosome–autosome interactions. Integrating single-cell multi-omics with sex-stratified cellular models will elucidate dosage–effect regulatory networks in adipocytes, hepatocytes, and other metabolically active cells, advancing targeted interventions for sex-specific metabolic disorders.

### 3.2. Sex Hormone Receptor Signaling Dynamics

Sex hormones are mainly secreted by the gonads. The ovaries of females mainly secrete estrogens and progesterone, while the testes of males mainly secrete androgens primarily composed of testosterone. These steroid hormones share a tetracyclic hydrocarbon backbone and are mainly derived from cholesterol. In detail, estrogen is characterized by an aromatic A-ring with a hydroxyl group at C3. Its synthesis begins with cholesterol conversion to pregnenolone via CYP11A1, followed by oxidation to progesterone. Progesterone undergoes sequential modifications by CYP17A1 to form androstenedione, which is reduced to testosterone. Testosterone is aromatized by aromatase, removing the C19 methyl group and inducing A-ring aromatization to yield estradiol. Estrogens and androgens, classically recognized for their roles in sexual differentiation and reproductive function, exert profound regulatory effects on systemic metabolic homeostasis. Accumulating evidence reveals striking sexual dimorphism in metabolic disease susceptibility mediated by these hormones. Epidemiological analyses demonstrate that premenopausal females exhibit preferential subcutaneous fat distribution and enhanced insulin sensitivity relative to males, conferring protection against visceral adiposity and metabolic dysfunction. This protective phenotype dissipates after menopause, with declining estrogen levels precipitating central obesity, cardiovascular pathologies, IR, and hypertension at rates paralleling those in male counterparts [27]. Conversely, clinical evidence suggests that hypogonadal males with reduced testosterone levels display elevated risks of abdominal obesity and T2DM, establishing a bidirectional relationship between androgen status and metabolic health [132,133,134]. These clinical observations, corroborated by preclinical studies, position sex hormones as pivotal modulators of HFD-induced metabolic adaptations and key determinants of sex-specific disease trajectories [135].

#### 3.2.1. Androgens/Androgen Receptor

Androgens, predominantly testosterone synthesized by testicular Leydig cells with minor adrenal contributions, regulate metabolic homeostasis through genomic and non-genomic mechanisms mediated by androgen receptors (ARs). The genomic pathway involves ligand-dependent AR nuclear translocation, subsequent binding to androgen response elements (AREs), and transcriptional regulation via coregulator recruitment [136]. Non-genomic actions rapidly modulate glucose and lipid metabolism through membrane-initiated signaling cascades. Preclinical models reveal that androgen deprivation or AR antagonism induces lipid dysregulation and IR [137], whereas AR activation (e.g., ostarine administration) counteracts metabolic perturbations in mice [138] Clinically, male hypogonadism strongly correlates with obesity, IR, and dyslipidemia, and these metabolic abnormalities are reversible through testosterone replacement therapy [139,140,141,142,143]. These findings collectively establish androgens/AR signaling as a central determinant of sex-specific metabolic responses.

The androgen–AR axis exerts dual regulatory control through central and peripheral pathways. Central modulation occurs via hypothalamic AR-mediated suppression of NF-κB-dependent protein tyrosine phosphatase 1B (PTP1B) expression, enhancing insulin sensitivity under HFD conditions [144]. Concurrently, AR signaling regulates energy balance by modulating pro-opiomelanocortin/neuropeptide Y (POMC/NPY) neuronal activity and attenuates neuroinflammation through TLR4/NF-κB/mitogen-activated protein kinase (MAPK) pathway inhibition, elevating anti-inflammatory cytokines IL-10/IL-13 [145]. In peripheral tissues, AR activation promotes skeletal muscle glucose uptake via glucose transporter 4 (GLUT4) induction [146], stimulates adipose lipolysis through hormone-sensitive lipase and LPL activation while suppressing lipogenesis via peroxisome proliferator-activated receptor gamma/stearoyl-CoA desaturase 1/acetyl-CoA carboxylase-1 downregulation, and enhances brown fat thermogenesis via UCP1 transcriptional activation [147]. Hepatic AR signaling improves lipid metabolism by increasing bile acid synthesis and taurine production via cysteine sulfinic acid decarboxylase regulation [148,149], whereas pancreatic AR activity enhances β-cell function through glucagon-like peptide-1 (GLP-1) receptor upregulation and focal adhesion kinase/phosphatidylinositol 3-kinase/mammalian target of rapamycin complex 2-mediated insulin secretion potentiation [150,151].

While females maintain lower circulating androgen levels, primarily through adrenal synthesis and peripheral aromatization, these hormones critically regulate reproductive development, bone homeostasis, and metabolic physiology. Pathologically, hyperandrogenism, a hallmark of polycystic ovary syndrome, exacerbates IR and T2DM risk in females [152,153]. Preclinical studies demonstrate that HFD-fed female rats with dihydrotestosterone supplementation develop profound IR, which can be reversed through hypothalamic-specific AR deletion [154]. Mechanistically, supraphysiological androgens disrupt metabolic homeostasis via dual pathways: centrally, they impair hypothalamic function by suppressing kisspeptin/gonadotropin-releasing hormone and neuronal activity while stimulating agouti-related peptide (AgRP) neurons to enhance feeding behavior, concurrently inducing leptin resistance and brown adipose tissue dysfunction [155,156]. Peripherally, androgen excess promotes visceral adiposity through adipocyte hypertrophy, macrophage infiltration, and oxidative stress while simultaneously inducing pancreatic β-cell damage and gut microbiota dysbiosis [154,157]. Notably, hepatic AR ablation in female mice attenuates HFD-induced metabolic derangements by preserving insulin signaling and mitigating weight gain [158].

Overall, the androgen–AR axis exhibits sexually dimorphic metabolic regulation. In males, physiological AR signaling maintains metabolic homeostasis through hypothalamic suppression of appetite via POMC/NPY neuronal modulation and NF-κB/PTP1B pathway inhibition, coupled with peripheral enhancements in skeletal muscle GLUT4-mediated glucose uptake, brown adipose UCP1-driven thermogenesis, and hepatic suppression of gluconeogenic enzymes. In contrast, female hyperandrogenism drives metabolic dysfunction via central appetite stimulation (AgRP upregulation/POMC suppression) and leptin resistance, compounded by peripheral visceral adipocyte inflammation, oxidative stress, and gut microbiota alterations that collectively exacerbate β-cell failure. These divergent pathways underscore the necessity for sex-specific therapeutic strategies targeting androgen/AR signaling in metabolic disease management.

#### 3.2.2. Estrogen and Estrogen Receptors

Estrogens, the predominant female sex hormone synthesized primarily by ovarian follicles and the corpus luteum with minor adrenal contributions, exerts critical regulatory effects on glucose and lipid metabolism through estrogen receptor α/β (ERα/ERβ)-mediated mechanisms. This signaling axis mitigates HFD-induced metabolic disturbances and protects against obesity-related pathologies [159,160]. Evidence from preclinical and clinical studies suggests that postmenopausal estrogen decline precipitates visceral adiposity, IR, atherogenic dyslipidemia, and cardiovascular risk elevation. These metabolic derangements are reversed through estrogen replacement therapy [161,162,163,164]. The estrogen/ER system orchestrates metabolic homeostasis via dual regulatory tiers: centrally, hypothalamic ERα engages glutamate/aspartate-rich carboxy-terminal domain-containing protein 1 (Cited1) in arcuate nucleus neurons to potentiate leptin sensitivity and suppress hyperphagia [165]. Peripherally, ER activation enhances skeletal muscle/adipose glucose uptake through PI3K–Akt–GLUT4 axis stimulation while preserving β-cell mass via anti-apoptotic mechanisms [52]. Concurrently, ERα modulates lipid metabolism by upregulating LDL receptor and scavenger receptor class B type I to accelerate cholesterol clearance [159,166,167,168] while suppressing visceral adipogenesis through AMP-activated protein kinase/peroxisome proliferator-activated receptor-α (AMPK/PPARα) pathway activation and adipose tissue inflammation resolution [29,169]. These coordinated actions establish estrogen signaling as a cornerstone of female metabolic health across the lifespan, with its age-related decline constituting a key driver of postmenopausal MetS susceptibility.

While males maintain lower physiological estrogen levels via adrenal synthesis and testicular aromatase-mediated androgen conversion, this hormone critically regulates bone homeostasis, reproductive function, and metabolic health through ERα/ERβ signaling. Estrogen modulates hypothalamic–pituitary–gonadal axis activity via negative feedback on luteinizing hormone/follicle-stimulating hormone secretion, while ERα activation suppresses visceral adiposity and enhances insulin sensitivity through adipocyte differentiation inhibition [170]. Clinical evidence from aromatase-deficient males reveals an estrogen-dependent MetS phenotype characterized by dyslipidemia, ectopic fat deposition, and glucose intolerance, which is reversed by estrogen therapy [170]. Preclinical studies further demonstrate that estrogen confers hepatic protection in male mice through formyl peptide receptor 2 induction, attenuating HFD-induced hepatocyte damage [171]. In summary, these findings underscore the conserved metabolic regulatory role of estrogen across sexes, with sex-specific circulating levels explaining divergent disease susceptibilities.

### 3.3. Genetic and Epigenetic Regulatory Networks

Epigenetic regulation dynamically modulates energy metabolism through DNA methylation and histone modifications at metabolic loci. The hypermethylation of the peroxisome proliferator-activated receptor gamma coactivator 1 α (PGC-1α) promoter suppresses mitochondrial function in cardiomyocytes, while HFD-induced tissue-specific alterations, including adipose PPARα hypomethylation and hepatic ERα histone hyperacetylation, disrupt lipid homeostasis and insulin signaling [172]. Pathological remodeling in metabolic diseases features β-cell insulin promoter hypermethylation and adipocyte PPARG hypermethylation, exacerbating glucose intolerance through impaired secretion and differentiation [173]. Concurrently, hepatic nuclear factor 4α methylation and enhancer of zeste homolog/tripartite motif-containing 14-mediated histone modifications drive diabetes susceptibility and inflammatory dysregulation [174,175].

Sexual dimorphism in epigenetic regulation profoundly shapes metabolic phenotypes. Genome-wide association studies reveal sex-biased obesity-risk loci: SAFB-like transcription modulator (SLTM) variants selectively increase male adiposity, whereas rare death inducer obliterator 1 (DIDO1) and *SLC12A5* mutations exhibit female-specific associations with elevated body mass index (>80% penetrance in females). Notably, clinical evidence indicated that DIDO1 variants correlate with hyperandrogenism and central adiposity in females, while solute carrier family 12 member 5 (SLC12A5*)* mutations heighten T2DM risk in this population [176]. X-chromosome epigenetic mechanisms further drive metabolic divergence. Xist-mediated chromatin remodeling calibrates metabolic plasticity in female mice [177], while male-specific hypermethylation of the X-linked angiotensin-converting enzyme 2 (ACE2) gene reduces pancreatic islet expression compared to that in females [178]. Key metabolic genes exhibit sex-divergent epigenetic controls: MC4R variants promote a male-biased preference for hypercaloric diets through dimorphic enhancer methylation [179], whereas fat mass and obesity-associated protein (FTO) polymorphisms differentially regulate iroquois homeobox 3/5 expression via AT-rich interaction domain 5B-dependent histone modifications, altering adipogenesis and thermogenic capacity [180].

These findings establish an integrative paradigm wherein sex-specific epigenetic networks encompassing X-chromosome reprogramming, allele-specific DNA methylation, and histone modifier activity sculpt metabolic dimorphism. The dynamic interplay between sex hormones and chromatin-modifying enzymes reinforces divergent regulatory trajectories, with androgen/estrogen signaling modulating epigenetic writer/eraser recruitment and X-linked modifiers like XIST fine-tuning metabolic adaptation. Future investigations should prioritize deciphering the molecular crosstalk between steroid receptors and epigenetic remodelers, elucidating X-chromosome dosage compensation mechanisms in metabolic regulation, and developing sex-stratified epigenetic therapies targeting MC4R, FTO, or ACE2 pathways. Such advances will catalyze precision medicine strategies for metabolic disorders through gender-informed chromatin modulation.

### 3.4. Gut Microbiota–Host Metabolic Crosstalk

The gut microbiota serves as a pivotal regulator of host metabolic homeostasis, with dysbiosis strongly implicated in obesity, IR, and MetS pathogenesis [181]. Preclinical evidence indicates that HFD induces microbial imbalance, characterized by an increased Firmicutes/Bacteroidetes (F/B) ratio, through the selective enrichment of Firmicutes over Bacteroidetes. This shift reduces populations of short-chain fatty acid (SCFA)-producing taxa (e.g., *Lachnospiraceae*, *Ruminococcus*), diminishing butyrate levels that normally reinforce intestinal barrier integrity via G-protein-coupled receptor 41/43 activation and histone deacetylase inhibition. The resultant barrier dysfunction facilitates systemic lipopolysaccharide translocation, driving chronic inflammation [182,183]. Concurrently, preclinical and clinical evidence shows that HFD-induced alterations in secondary bile acid metabolism exacerbate hepatic lipid accumulation and suppress GLP-1 secretion through farnesoid X receptor (FXR) signaling activation [184,185].

Sexual dimorphism profoundly influences HFD-induced microbiota remodeling. Preclinical evidence indicates that male mice develop adipose inflammation and IR associated with *Bacteroidaceae*/*Clostridia* expansion alongside *Lactobacillus* and *Desulfovibrionaceae* enrichment [11,186,187]. In contrast, female counterparts exhibit more severe intestinal barrier disruption, evidenced by tight junction protein dysregulation and colonic proinflammatory cytokine elevation, correlating with *Actinomycetota*/*Bacillota* depletion [11,183,187]. Baseline microbial differences underlie this divergence: males harbor lower α-diversity with *Prevotella*/*Veillonella* predominance, while females maintain richer communities enriched in SCFA producers [10,188,189,190,191,192,193,194]. These sex-specific communities differentially modulate host metabolism through SCFA-mediated histone deacetylase inhibition/G-protein-coupled receptor signaling and bile acid–FXR interactions that regulate hepatic lipid synthesis genes (e.g., SREBP-1c). Emerging evidence further indicates the promise of microbiota–gut–brain axis signaling, where SCFAs suppress hypothalamic NPY via vagal afferents to reduce hyperphagia [184].

These findings establish gut microbiota sexual dimorphism as a central determinant of HFD-induced metabolic disparities. While male susceptibility stems from low-diversity baseline flora predisposing individuals to proinflammatory taxon shifts, female pathophysiology involves α-diversity loss compromising barrier-protective SCFA producers. Mechanistically, sex-specific microbiota metabolites engage complementary pathways. For instance, SCFAs enhance insulin sensitivity through epigenetic and receptor-mediated mechanisms, while bile acids modulate lipid homeostasis via FXR. Critical knowledge gaps persist regarding sex hormone–microbiota crosstalk and translational applicability to human populations. Future investigations should integrate multi-omics profiling with gnotobiotic models to decipher dynamic microbiota–host interactions, enabling the development of sex-stratified therapeutic strategies targeting microbial networks for the precision management of metabolic disorders.

### 3.5. Neuroendocrine Pathway Dimorphisms

Hypothalamic glial cells play pivotal roles in energy homeostasis and metabolic regulation. In HFD-fed mice, hypothalamic microglia exhibit rapid activation marked by morphological changes and elevated proinflammatory cytokine production. These microglia undergo immunometabolic reprogramming, shifting toward fatty acid oxidation and oxidative phosphorylation with suppressed glycolysis, while displaying mixed M1/M2-like polarization [195]. HFD further induces astrocyte proliferation and NF-κB-dependent IL-6 secretion [195], with notable sex-specific responses: male mice develop hypothalamic inflammation under HFD, whereas females remain resistant [196]. This protection may be attributed to the function of Cited1, which is enriched in hypothalamic POMC neurons and governs appetite and energy balance by facilitating crosstalk between ERα and leptin-activated Stat3 [165]. Preclinical models demonstrate that POMC neuron-specific cited1 deletion causes diet-induced obesity exclusively in females [165].

Collectively, sex-specific neuroendocrine pathways engage distinct molecular hubs: serotonin 2C receptor helps maintain metabolic stability in males, while Cited1 orchestrates hormone–neuron interactions in females. Future research directions include further investigations into the differences in the responses of other hypothalamic neuron populations to HFD exposure across sexes. Combining the multidimensional information of patients’ genetic backgrounds, lifestyles, and disease status, we can design a personalized treatment plan to precisely regulate the neuroendocrine pathway in order to improve metabolic status.

Interestingly, increasing evidence suggests that maternal HFD adherence leads to obesity, IR, impaired glucose tolerance, and increased cardiovascular disease risk in offspring. The sex-specific remodeling of hypothalamic energy homeostasis networks represents a critical neural mechanism underlying this detrimental impact of maternal HFD consumption on offspring. Preclinical studies demonstrate that maternal HFD exposure sex-dependently reprograms arcuate nucleus circuitry: Male offspring exhibit amplified orexigenic signaling through increased AgRP neuron populations, elevated AgRP-to-POMC neuron ratios, and expanded arginine vasopressin/retinoic acid receptor-related orphan receptor β (Avp/Rorb) neuronal clusters, collectively driving hyperphagia and insulin dysregulation that predisposes them to obesity [177]. Neuron–glial interaction analyses reveal the male-specific enhancement of AgRP neuron–astrocyte crosstalk, coupled with the activation of neurotrophic regulatory networks involving the obesity-associated gene neuronal growth regulator 1. In contrast, female offspring show compensatory upregulation of the X-chromosome-encoded long noncoding RNA Xist in hypothalamic tissue, suggesting sex chromosome-linked epigenetic mechanisms may mitigate metabolic perturbations caused by maternal overnutrition [177]. Additionally, maternal HFD leads to metabolic disorder in offspring through the upregulation of cytokines, neutrophil infiltration, damage to the intestinal barrier, and the induction of systemic inflammation. Moreover, maternal HFD impairs pancreatic β-cell function in offspring; significantly increases lipid content, blocks the AMPK/SIRT1/PGC-1α signaling pathway; and thereby promotes the occurrence of obesity, hyperglycemia, and cardiovascular diseases [197]. These findings establish a molecular framework wherein maternal metabolic stress induces sexually dimorphic hypothalamic rewiring through divergent neuron–glia interaction patterns and neuroendocrine pathway activation, ultimately shaping offspring susceptibility to energy imbalance.

## 4. Dietary and Pharmacotherapeutic Interventions Targeting Sex-Specific HFD-Induced Metabolic Diseases

Sexual dimorphism profoundly influences dietary regulation and pharmacological responses through divergent nutritional requirements and metabolic adaptations. Females benefit from high-fiber diets that optimize gut microbiota and modulate estrogen levels to reduce breast cancer risk [198], whereas males require restricted refined carbohydrate intake to mitigate MetS predisposition [199]. While protein consumption enhances energy metabolism in both sexes, females exhibit superior muscle anabolic responses [200,201]. Lipid regulation strategies also diverge: clinical data show that omega-3 and monounsaturated fatty acids preferentially support female cardiovascular health [202], while males require focused monounsaturated fat intake for adiposity control [203]. The Mediterranean diet exerts stronger cardioprotective effects on males via anti-inflammatory mechanisms [204], whereas ketogenic diets may compromise female gut barrier integrity despite short-term weight loss [205]. Micronutrient antioxidants (e.g., vitamins C/E, selenium, zinc) alleviate HFD-induced metabolic stress by scavenging reactive oxygen species and stabilizing hormonal balance [206,207], with iron supplementation being particularly critical for female physiology [208].

Pharmacological interventions exhibit sex-biased efficacy and safety profiles. Clinical data reveal that reduced cytochrome P450 activity predisposes females to elevated drug concentrations and adverse effects [209]. Glucose-lowering agents demonstrate sex-divergent responses: males show heightened sulfonylurea sensitivity [210], while GLP-1 receptor agonists (e.g., liraglutide) elicit stronger weight-modulatory effects on females [211]. Fluctuating estrogen levels during menstrual cycles and menopause alter metformin’s cardioprotective efficacy [212,213,214], whereas males exhibit superior metabolic responses to sodium-glucose co-transporter 2 inhibitors and statins [215]. Hormone replacement strategies further underscore sexual dimorphism: clinical studies show that estrogen therapy enhances insulin sensitivity in postmenopausal females [216], while testosterone supplementation reduces visceral adiposity and improves IR in males [217,218,219]. However, long-term safety profiles require rigorous sex-specific evaluation [97].

These findings underscore the necessity for sex-tailored therapeutic approaches. Nutritional strategies should prioritize female gut microbiota optimization through fiber-rich diets and targeted micronutrients, while males require carbohydrate restriction and lipid profile modulation. Pharmacological regimens must account for sex-specific drug metabolism patterns and hormonal interactions, with individualized adjustments based on metabolic risk profiles and therapeutic objectives.

## 5. Sex Differences in Exercise-Related Exercise Capacity and Metabolic Health

Sexual dimorphism in exercise-mediated metabolic regulation arises from fundamental differences in skeletal muscle architecture and energy flux pathways. Males exhibit higher lean mass and glycolytic type II fiber predominance, favoring anaerobic performance but exhibiting lower basal energy expenditure compared to females [200,220]. Conversely, female muscle physiology prioritizes oxidative metabolism through type I fiber dominance and estrogen–AMPK axis-driven lipid mobilization, particularly enhancing fatty acid oxidation during endurance activities [221,222,223]. This metabolic divergence manifests as male superiority in power/strength tasks versus female advantages in fatigue resistance and fat utilization efficiency [222,224]. Hormonal regulation further stratifies these responses: males display greater exercise-induced AMPK activation [225], while females optimize intramuscular lipid storage and hypoxic metabolic flexibility via 17β-estradiol signaling [223,226].

Exercise intervention outcomes demonstrate sex-biased cardiometabolic benefits. High-intensity interval training preferentially enhances male glycolytic capacity and VO_2_max, albeit with paradoxical reductions in insulin sensitivity [200,227,228]. Females achieve superior cardioprotection and all-cause mortality reduction through moderate-intensity regimens [229]. Sexual dimorphism extends to HFD-induced metabolic rescue: preclinical studies have shown that male mice develop exercise-aggravated cardiac lipotoxicity under high-intensity interval training [230], while females improve insulin sensitivity via endurance training through IRS-1 serine dephosphorylation and mitochondrial biogenesis [200]. Menstrual cycle dynamics introduce additional complexity: luteal-phase progesterone elevation suppresses lipid oxidation while enhancing glucose utilization, necessitating hormone-phase-adapted exercise prescriptions [200]. Notably, sex-specific intergenerational metabolic programming emerges where maternal preconception/prenatal exercise preferentially potentiates male offspring metabolic fitness [231].

These findings establish a paradigm wherein sex-specific exercise adaptations are orchestrated through intersecting hormonal, cellular, and metabolic networks. While male physiology capitalizes on anabolic capacity and acute AMPK responsiveness, female metabolic resilience stems from estrogen-optimized oxidative efficiency and lipid flexibility. Critical knowledge gaps persist regarding sex hormone–exercise crosstalk at a molecular resolution and the clinical translation of sex-stratified exercise therapies. Future research must employ multi-omics approaches to decode the dynamic interactome of exercise–sex hormone–metabolism networks, enabling precision exercise regimens tailored to hormonal status and metabolic phenotypes.

## 6. Conclusions and Perspectives

The pervasive sexual dimorphism in HFD-induced metabolic disorders underscores an urgent need to recalibrate biomedical research paradigms historically skewed toward male physiology. This synthesis reveals that sex-specific metabolic vulnerabilities emerge through an intricate interplay of sex chromosome-mediated transcriptional programs, steroid hormone-regulated energy partitioning, dynamic epigenetic remodeling, microbiota–metabolite crosstalk, and hypothalamic nutrient-sensing plasticity. Crucially, the sexually divergent therapeutic responses observed in exercise adaptation and drug metabolism underscore the limitations of one-size-fits-all interventions, advocating for clinical strategies that harmonize sex-specific molecular signatures with physiological traits. In addition, other variables, such as age or ethnicity, may also play a role in the sex-dependent impact of HFD on metabolic diseases, which warrants further studies in the future.

Looking ahead, the field must prioritize elucidating the spatiotemporal choreography of sex hormone fluctuations across developmental milestones and their crosstalk with epigenetic–microbiome networks. Advanced multi-omics integration spanning metabolomic, epigenomic, and proteomic landscapes will be instrumental in building predictive models that map sex-specific molecular cascades to clinical phenotypes. Parallel efforts should focus on translating these insights into precision medicine through two synergistic pathways: developing therapies targeting conserved nodes of sexually dimorphic regulation and implementing personalized prevention frameworks that account for hormonal status, genetic predisposition, and environmental exposures. By bridging these frontiers, we can catalyze a transformative shift toward sex-stratified therapeutic regimens, ultimately enabling tailored nutritional and pharmacological strategies to combat HFD-associated cardiometabolic diseases with precision.

## Figures and Tables

**Figure 1 ijms-26-04777-f001:**
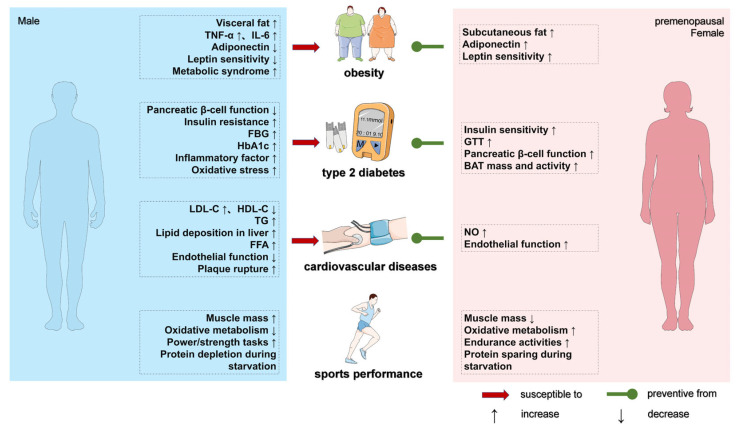
Sex differences in metabolic responses associated with HFD and the development of metabolic diseases. Some elements of this figure are from Servier Medical Art (https://smart.servier.com/, accessed on 15 of March 2025).

**Figure 2 ijms-26-04777-f002:**
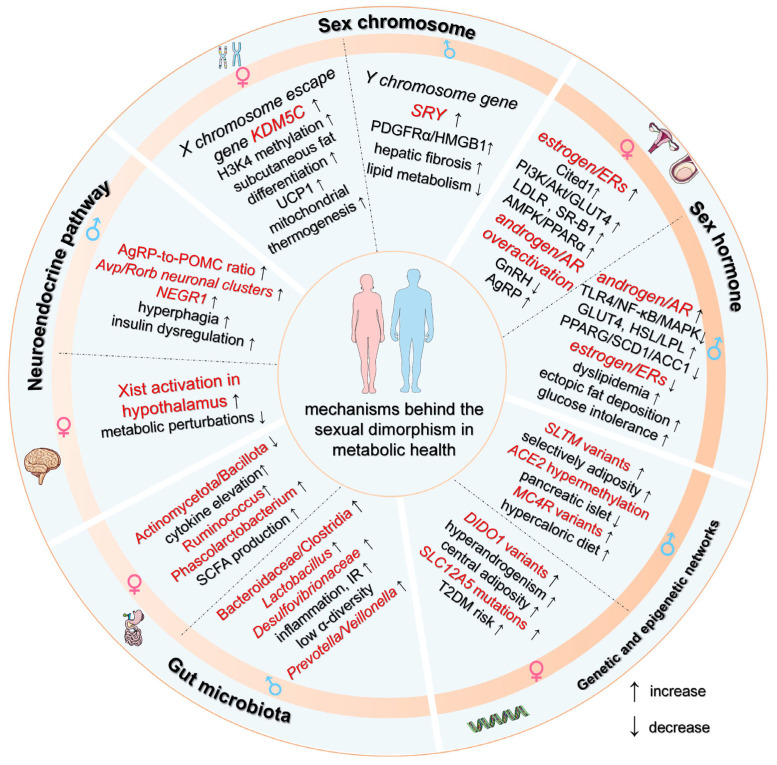
Mechanisms behind sex differences in the development of metabolic diseases. Notes: Some elements in this figure are from Servier Medical Art (https://smart.servier.com/, accessed on 15 of March 2025). KDM5C: lysine demethylase 5C; UCP1: uncoupling protein 1; SRY: sex-determining region Y; ERs: estrogen receptors; PDGFRα: platelet-derived growth factor receptor α; HMGB1: high-mobility group box-1 protein; Cited1: glutamate/aspartate-rich carboxy-terminal domain-containing protein 1; PIK3: phosphatidylinositol 3-kinase; Akt: protein kinase b; GLUT4: glucose transporter type 4; LDLR: low-density lipoprotein receptor; SR-B1: scavenger receptor class B type 1; AMPK: adenosine 5′-monophosphate (AMP)-activated protein kinase; PPARα: peroxisome proliferator-activated receptor α; AR: androgen receptor; GnRH: gonadotropin-releasing hormone; AgRP: agouti-related protein; TLR4: toll-like receptor 4; NF-κB: nuclear factor-kappa B; MAPK: mitogen-activated protein kinase; HSL: hormone-sensitive triglyceride lipase; LPL: lipoprotein lipase; PPARG: peroxisome proliferator-activated receptor gamma; SCD1: stearoyl-CoA desaturase 1; ACC1: acetyl-CoA carboxylases-1; NEGR1: neuronal growth regulator 1.

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
