# Peer review of "High-Fat Diet and Metabolic Diseases: A Comparative Analysis of Sex-Dependent Responses and Mechanisms"

_ijms, 2025, doi:10.3390/ijms26104777_

Round 1
Reviewer 1 Report
Comments and Suggestions for Authors
This is a timely and comprehensive narrative review on sex-specific responses to high-fat diet and related chronic diseases. The topic is highly relevant and aligns well with current efforts toward precision medicine and sex-informed research. The manuscript is well-organized, scientifically sound, and supported by recent literature. It provides important insights into mechanisms of metabolic dimorphism.
Positive points:
It is a complete and up-to-date revison of sex-dependent mechanisms of compensatory responses to high-fat diet and related chronic diseases.
The text is excellently organized with a clear structure. The manuscript is well written. The information progressively adds knowledge.
Figures are well-designed and well-correlated with the text.The figures are self-explanatory and contain a summary of the relevant information. The legends are very descriptive.
Suggestions for improvement:
Although this is a narrative review, the authors should briefly describe the literature search strategy (e.g., databases consulted, inclusion/exclusion criteria) to enhance understanding and reproducibility.
I suggest a more critical and concise approach, highlighting the convergent points, but emphasizing the controversies and obscure points of knowledge. I believe this will bring more robustness and impact to the article.
If it is possible, in order to add more value to this manuscript, I also suggested adding some information about the correlation between sex and other variables, such as age, pregnancy, menopause, or ethnicity.
Author Response
Response to Reviewer #1
General comments
This is a timely and comprehensive narrative review on sex-specific responses to high-fat diet and related chronic diseases. The topic is highly relevant and aligns well with current efforts toward precision medicine and sex-informed research. The manuscript is well-organized, scientifically sound, and supported by recent literature. It provides important insights into mechanisms of metabolic dimorphism.
Response:
We appreciate your positive feedback and valuable suggestions. We have made the necessary revisions to the manuscript based on your comments.
Specific concerns
SC#1:Although this is a narrative review, the authors should briefly describe the literature search strategy (e.g., databases consulted, inclusion/exclusion criteria) to enhance understanding and reproducibility.
Response SC#1:
Thanks for your suggestion. We agree that the provision of literature search strategies is of great significance for enhancing understanding and reproducibility. A brief discussion of literature search strategy has been included in the revised manuscript (lines 70-79).
SC#2:I suggest a more critical and concise approach, highlighting the convergent points, but emphasizing the controversies and obscure points of knowledge. I believe this will bring more robustness and impact to the article.
Response SC#2:
Thank you for your valuable suggestions. We have carefully refined the manuscript to strengthen critical analysis and conciseness, particularly in the Section 2. Redundant statements were streamlined across the text to enhance focus. While we maintained the original structure to preserve coherence, these adjustments clarify unresolved issues and sharpen the paper's critical tone.
SC#3:If it is possible, in order to add more value to this manuscript, I also suggested adding some information about the correlation between sex and other variables, such as age, pregnancy, menopause, or ethnicity.
Response SC#3:
Thank you for the insightful suggestion to explore correlations between sex and additional variables such as age, pregnancy, menopause, and ethnicity. In the revised manuscript, we have expanded the discussion of sex-dependent metabolic effects in females, particularly highlighting pregnancy and menstrual cycle-related interactions with HFD outcomes.
Regarding age and ethnicity, while preclinical evidence suggests potential interactions. (e.g., older male mice exhibit stronger HFD-induced inflammatory responses compared to younger females [1], and middle-aged male mice show distinct metabolic adaptations to time-restricted HFD [2]), the mechanistic underpinnings of these age-sex relationships remain poorly characterized in the literature.
Similarly, ethnicity-mediated differences in sex-specific HFD responses are severely understudied across both preclinical and clinical research. To maintain the clarity and focus of our core argument on sex hormone-driven mechanisms, we have refrained from speculative discussions on age or ethnicity in the main text. However, we now explicitly acknowledge these gaps as critical future research directions in the Conclusions and Perspectives section (lines 708-710).
References:
[1] Evans AK, Saw NL, Woods CE, et al. Impact of high-fat diet on cognitive behavior and central and systemic inflammation with aging and sex differences in mice. Brain Behav Immun. 2024;118:334-354. doi:10.1016/j.bbi.2024.02.025]
[2] Chaix A, Deota S, Bhardwaj R, Lin T, Panda S. Sex- and age-dependent outcomes of 9-hour time-restricted feeding of a Western high-fat high-sucrose diet in C57BL/6J mice. Cell Rep. 2021;36(7):109543. doi:10.1016/j.celrep.2021.109543]

Reviewer 2 Report
Comments and Suggestions for Authors
The authors reviewed current evidence on HFD-induced metabolic dysregulation through a sex-specific perspective, elucidating underlying mechanisms spanning molecular pathways to clinical manifestations.. This Review article carries novel information. Overall, this article is well written and can be published after a few minor revisions.
My comments are as follows:
- Title: HFD on humans or animals? It should be called western diet in humans. “Chronic diseases” is inadequate and incompatible with the content (metabolic diseases?).
- Section 1: The authors mixed the evidence of preclinical and clinical evidences. It wound be a plus if the authors can organize better in this point to benefit the readers for further researches. Same efforts should be made for the following sections. For example, “HFD induces microbial imbalance characterized by increased Firmicutes/Bacteroidetes (F/B) ratio through selective enrichment of Firmicutes over Bacteroidetes” is unclear whether this is a clinical or preclinical study (section 2.4).
- Section 2.5: Maternal HFD is another topic and should be discussed separately and clearly.
- 1.: The content of postmenopausal female is missing.
- Acronyms should be used correctly. Examples are HFD (abstract) NF-kB (page 1), etc.... In contrast, POMC, GRK2, and MDA are not necessary, etc...
- A few typos or editing errors: Title of 2.3 should not be in bold font.
- Reference styles should be consistent. For example, reference 2 is different from reference 3.
Author Response
Point-by-Point Response to Reviewers’ Comments
Dear editor and reviewers,
Thanks for your helpful comments and suggestions. We have seriously considered all the concerns and carefully revised the manuscript accordingly. Revisions are highlighted in red in the revised manuscript. We believe that the quality of the manuscript has been significantly improved with these revisions based on the reviewers’ suggestions. The point-by-point responses to reviewers’ concerns are as followings:
Response to Reviewer #2
General comments:
The authors reviewed current evidence on HFD-induced metabolic dysregulation through a sex-specific perspective, elucidating underlying mechanisms spanning molecular pathways to clinical manifestations. This Review article carries novel information. Overall, this article is well written and can be published after a few minor revisions.
Response:
We appreciate your positive feedback and valuable suggestions. We have made the necessary revisions in the revised manuscript based on your comments.
Specific concerns
SC#1:Title: HFD on humans or animals? It should be called western diet in humans. “Chronic diseases” is inadequate and incompatible with the content (metabolic diseases?).
Response SC#1:
Thank you for raising this important point regarding terminology. We fully agree that precise language is critical in scientific communication. While "western diet" is a well-established term in human studies, it typically encompasses multiple dietary components, including high sugar, processed foods, and refined carbohydrates, in addition to elevated fat intake. In contrast, the clinical studies cited in our review specifically investigated dietary interventions where high-fat composition was the primary variable under examination. Thus, "HFD" (high-fat diet) was intentionally selected to align with the mechanistic focus of these studies.
We sincerely appreciate your suggestion to use the term “metabolic diseases”, which aligns more precisely with the content. We have made this change in the revised manuscript.
SC#2:Section 1: The authors mixed the evidence of preclinical and clinical evidences. It wound be a plus if the authors can organize better in this point to benefit the readers for further researches. Same efforts should be made for the following sections. For example, “HFD induces microbial imbalance characterized by increased Firmicutes/Bacteroidetes (F/B) ratio through selective enrichment of Firmicutes over Bacteroidetes” is unclear whether this is a clinical or preclinical study (section 2.4).
Response SC#2:
We appreciate your astute observation. In the revised manuscript, we have rigorously clarified the distinction between preclinical and clinical evidence by explicitly labeling the study type (e.g., “clinical”, “preclinical”) and specifying the research subjects (e.g., “rodents”, “human”) within relevant citations. For instance, in Section 2.4, the statement on HFD-induced microbial imbalance now reads: "Preclinical evidence indicates that HFD induces microbial imbalance characterized by increased Firmicutes/Bacteroidetes (F/B) ratio through selective enrichment of Firmicutes over Bacteroidetes)” (lines 527-530). Similar modifications have been applied throughout the text to ensure consistency.
SC#3:Section 2.5: Maternal HFD is another topic and should be discussed separately and clearly.
Response SC#3:
Thank you for highlighting the need to clarify this aspect. In the revised Section 2.5, we have added a dedicated paragraph to explicitly address maternal HFD as a distinct topic. This section now separately discusses (1) sex-specific differences in offspring metabolic outcomes, (2) mechanisms from Neuroendocrine pathway and inflammation related changes (lines 587-607).
SC#4:The content of postmenopausal female is missing.
Response SC#4:
Thank you for emphasizing the importance of postmenopausal metabolic health. In the revised manuscript, we have strengthened the discussion of postmenopausal females by explicitly labeling their unique physiological context in relevant sections. For example, Section 1 (e.g., lines 95-98; 112-115; 210-214) and section 2.2.2. summarized the metabolic changes and its underlying molecular mechanisms in postmenopausal women. These refinements systematically address postmenopausal physiology within the existing framework.
SC#5:Acronyms should be used correctly. Examples are HFD (abstract) NF-kB (page 1), etc.... In contrast, POMC, GRK2, and MDA are not necessary, etc...
Response SC#5:
Thank you very much for your meticulous review and suggestions. Regarding the issue you mentioned about "inappropriate use of abbreviations", we have carefully checked and revised it. Specifically, we have added the abbreviations of HFD and NF-κB when they firstly appeared. Additionally, we have removed the abbreviations of words that appeared only once (e.g., GRK2, MDA, CaSR, and vWAT) in the text to ensure the correct and necessary use of abbreviations as suggested.
SC#6:A few typos or editing errors: Title of 2.3 should not be in bold font.
Response SC#6:
Thank you for highlighting the formatting inconsistency. We have meticulously standardized the section title font in accordance with journal guidelines and conducted a full-text audit to cross-verify all headings, subheadings, and labels. Formatting consistency is now fully maintained throughout the manuscript.
SC#7:Reference styles should be consistent. For example, reference 2 is different from reference 3
Response SC#7:
Thank you for noting this inconsistency. We have systematically reviewed and standardized the reference formatting to align with journal guidelines, ensuring full consistency across the manuscript.

Reviewer 3 Report
Comments and Suggestions for Authors
- the authors provide a paragraph on the chemical structure and synthesis of estrogen - progesterone and testosterone.
- the authors should provide the definition of the insulin resistance and the associated pathologies.
-the authors should provide a clear definition of metabolic syndrome
- the authors should provide a paragraph on the cerebrovascular disease and cardiovascular disease and gender differences (see Regitz-Zagrosek and Gebhard, Gender medicine: effects of sex and gender on cardiovascular disease manifestation and outcomes, Nat Rev Cardiol, 2023 (4):236-247. doi: 10.1038/s41569-022-00797-4.
And Kathryn M Rexrode et al., The Impact of Sex and Gender on Stroke, Circ Res. 2022, 130(4):512-528. doi: 10.1161/CIRCRESAHA.121.319915.
- the authors should define and discuss the Klinefelter syndrome.
Author Response
Point-by-Point Response to Reviewers’ Comments
Dear editor and reviewers,
Thanks for your helpful comments and suggestions. We have seriously considered all the concerns and carefully revised the manuscript accordingly. Revisions are highlighted in red in the revised manuscript. We believe that the quality of the manuscript has been significantly improved with these revisions based on the reviewers’ suggestions. The point-by-point responses to reviewers’ concerns are as followings:
Response to Reviewer #3
Specific concerns
SC#1: the authors provide a paragraph on the chemical structure and synthesis of estrogen - progesterone and testosterone.
Response SC#1:
We appreciate your valuable suggestions. We have added a brief paragraph on the chemical structure and synthesis of estrogen, progesterone and testosterone in the manuscript based on your comments (lines 361-369).
SC#2: the authors should provide the definition of the insulin resistance and the associated pathologies.
Response SC#2:
Thanks for your valuable suggestion. We have included a brief introduction of insulin resistance and the associated pathologies in the revised manuscript (lines 128-134).
SC#3:the authors should provide a clear definition of metabolic syndrome
Response SC#3:
Thank you for your thoughtful reminder. We have added the definition of metabolic syndrome and cited a related reference in the revised manuscript (lines 102-105).
reference:
Bovolini A, Garcia J, Andrade MA, Duarte JA. Metabolic Syndrome Pathophysiology and Predisposing Factors. Int J Sports Med. 2021;42(3):199-214. doi:10.1055/a-1263-0898.
SC#4: the authors should provide a paragraph on the cerebrovascular disease and cardiovascular disease and gender differences.
Response SC#4:
We appreciate your constructive suggestion. We have read the articles you recommended and found them helpful in improving our manuscript. We have cited these two articles and a paragraph on the cerebrovascular disease and gender differences has been added in the revised manuscript (lines 215-228).
reference:
- Regitz-Zagrosek, V.; Gebhard, C., Gender medicine: effects of sex and gender on cardiovascular disease manifestation and outcomes. Nat Rev Cardiol 2023, 20, (4), 236-247.
- Rexrode, K. M.; Madsen, T. E.; Yu, A. Y. X.; Carcel, C.; Lichtman, J. H.; Miller, E. C., The Impact of Sex and Gender on Stroke. Circ Res 2022, 130, (4), 512-528.
SC#5:the authors should define and discuss the Klinefelter syndrome.
Response SC#5:
A brief definition and discussion of Klinefelter syndrome has been included in the revised manuscript (lines 325-331). Thanks for your suggestion.
